# Experimental Analysis of Deep-Sea AUV Based on Multi-Sensor Integrated Navigation and Positioning

**Yixu Liu** [1,2], **Yongfu Sun** [1,*], **Baogang Li** [1,3], **Xiangxin Wang** [1,4] and **Lei Yang** [1]

1. National Deep-Sea Base Management Center, Qingdao 266237, China; lyx@ndsc.org.cn (Y.L.); lbg@ndsc.org.cn (B.L.); wangxiangxin@ndsc.org.cn (X.W.); yangsir@ndsc.org.cn (L.Y.)
2. College of Environmental Science and Engineering, Ocean University of China, Qingdao 266101, China
3. Logistics Engineering College of SMU, Shanghai Maritime University, Shanghai 200135, China
4. College of Intelligent Systems Science and Engineering, Harbin Engineering University, Harbin 150001, China
* Correspondence: sunyongfu@ndsc.org.cn; Tel.: +86-137-0898-8003

**Abstract:** The operation of underwater vehicles in deep waters is a very challenging task. The use of AUVs (Autonomous Underwater Vehicles) is the preferred option for underwater exploration activities. They can be autonomously navigated and controlled in real time underwater, which is only possible with precise spatio-temporal information. Navigation and positioning systems based on LBL (Long-Baseline) or USBL (Ultra-Short-Baseline) systems have their own characteristics, so the choice of system is based on the specific application scenario. However, comparative experiments on AUV navigation and positioning under both systems are rarely conducted, especially in the deep sea. This study describes navigation and positioning experiments on AUVs in deep-sea scenarios and compares the accuracy of the USBL and LBL/SINS (Strap-Down Inertial Navigation System)/DVL (Doppler Velocity Log) modes. In practice, the accuracy of the USBL positioning mode is higher when the AUV is within a 60° observation range below the ship; when the AUV is far away from the ship, the positioning accuracy decreases with increasing range and observation angle, i.e., the positioning error reaches 80 m at 4000 m depth. The navigational accuracy inside and outside the datum array is high when using the LBL/SINS/DVL mode; if the AUV is far from the datum array when climbing to the surface, the LBL cannot provide accurate position calibration while the DVL fails, resulting in large deviations in the SINS results. In summary, the use of multi-sensor combination navigation schemes is beneficial, and accurate position information acquisition should be based on the demand and cost, while other factors should also be comprehensively considered; this paper proposes the use of the LBL/SINS/DVL system scheme.

**Keywords:** multi-sensor integrated navigation; underwater positioning; deep sea; autonomous underwater vehicle

## 1. Introduction

With rapid socio-economic, scientific and technological development and population growth, the problems of living space limitations, natural resources and disputes over rights and interests on land, where human beings have lived for a long time, have begun to come to the fore. This has led us to explore the development of the oceans, which cover almost 71% of the Earth's surface [1,2]. Rich in natural resources such as organisms, minerals, oil and natural gas, the oceans have become the focus of attention for countries around the world. Marine scientific research, marine environmental monitoring, deep-sea resource development and other marine activities are becoming increasingly common [3]. These marine activities require underwater vehicles as carrier platforms, such as AUVs. The world's most powerful maritime countries are building their own national PNT (Positioning, Navigation, Timing) systems [4,5], which is a systematic project involving the integration of land, sea, air and sky and can provide high-precision spatial and temporal information services to users throughout the domain.

To navigate safely underwater, submersibles need "eye navigation". This can provide information such as the position, velocity and attitude for the submarine, to help it to move underwater [6]. Inaccurate navigational information can make it difficult to accurately map the topography of the seabed [7]. To ensure the successful completion of underwater missions and to obtain relatively accurate underwater measurement data, it is necessary to have long-term autonomous high-precision underwater navigation and positioning capabilities and to be stealthy.

It is difficult to realize high-precision navigation and positioning by relying on the navigation and positioning information obtained from a single sensor. In order to meet the navigation requirements, SINS and DVL are usually used in combination [8]. The SINS/DVL integrated navigation system is used for the underwater navigation and positioning of the AUV. However, when using the ship position projection method for positioning, the positioning error will accumulate over time [9]. After years of development, GNSS (Global Navigation Satellite System) technology can achieve positioning accuracy in the decimeter or even centimeter range, providing highly accurate position information above the water surface [10–12]. The joint GNSS-A (acoustic) technology can provide a spatial and temporal reference for the establishment of ocean geodetic references and underwater positioning [13,14]. When the AUV is submerged for a period of time, the surface GNSS is used to correct the accumulated errors. In this scheme, the AUV has to travel continuously between the underwater operation site and the surface [15]. This not only reduces the operational efficiency and increases energy consumption, but also makes the AUV's position more vulnerable to detection.

Due to the rapid attenuation of electromagnetic wave energy in water, the energy attenuation becomes more pronounced as the distance increases. The good propagation characteristics of acoustic signals in water mean that underwater acoustic positioning systems are currently the main method of underwater absolute position transmission [16]. The most commonly used underwater acoustic positioning systems are mainly USBL and LBL [17]. USBL is characterized by its ease of use, flexibility and cost-effectiveness, but the accuracy is not very high. USBL combined with other navigation schemes has become a research hotspot in underwater navigation [18–20]. The main advantage of LBL is its high accuracy, but there is the problem of its high cost of use [21]. The ocean is a complex hydrodynamic environment where environmental noise and various errors interact to reduce the positioning accuracy. Scholars from various countries have performed research on the modeling of the sound velocity error and acoustic tracing correction methods in high-precision underwater positioning [22–24].

Once an accurate position result has been obtained, whether from the LBL or USBL, the AUV can be navigated using a combination of sensors. Liu et al. proposed an underwater AUV navigation and positioning algorithm using an LBL acoustic positioning system, an ADCP (Acoustic Doppler Current Profiler) and a depth meter assisting INS [25]. Miller, P. A. et al. proposed a tightly integrated system based on LBL/DVL/INS. Zhang et al. put forward an underwater positioning algorithm based on the interactive assistance of a SINS and LBL, and this algorithm mainly includes an optimal correlation algorithm with the aided tracking of an SINS/DVL/magnetic compass pilot (MCP), a three-dimensional T-DOA positioning algorithm with Taylor series expansion and a multi-sensor information fusion algorithm [26].

There are now many countries with mature AUV products that have navigation methods adapted to their operational scenarios. For example, Bluefin-21 uses a high-precision SINS as its primary navigation method, achieving an imputed drift rate of less than 0.1% of range, and is also equipped with a DVL, SVS (Sound Velocity Sensor) and GNSS. To complement the SINS, it is also equipped with USBL to further ensure accurate navigation and positioning [27]. The AUV of China's "Diving Dragon III" adopts the combined navigation mode of SINS and DVL when operating near the bottom and simultaneously fuses the USBL positioning information and depth information. With this combination of navigation modes, it can navigate along the set survey line, and

the positioning accuracy is better than 3‰ × D (D is the near-bottom voyage of the AUV), which can meet the needs of AUV near-bottom fine terrain sweeping. As the first manned submersible to dive to 7000 m underwater, "Jiaolong" uses a combination of USBL, navigation position projection navigation and DVL for navigation and positioning in near-bottom operation [28].

Based on knowledge from disciplines such as physical oceanography and ocean mapping, we know that AUVs are significantly more difficult to navigate and position in deep-water scenarios than in shallow water. This also means that some simple navigation and positioning modes may no longer be applicable. There has been little discussion of the simultaneous use of both the USBL and LBL modes on an AUV, especially in deep-sea scenarios. Based on the above discussion, this paper presents an experimental analysis of the navigation and positioning of a deep-sea AUV. For the same AUV, comparative experiments were conducted using two navigation and positioning modes to analyze the accuracy of USBL and LBL in position calibration.

## 2. Methodology

### 2.1. Underwater Acoustic Positioning System

GNSS-A is a technology that combines GNSS technology and hydroacoustic positioning technology to transmit spatial reference information to fixed underwater datum points or mobile submersibles. Acoustic positioning technology is the best way to transmit acoustic signals, and the most commonly used products are LBL and USBL [13].

The LBL system adopts the "ask-answer" mode, which uses spherical rendezvous to solve the geodetic coordinates of the underwater vehicle by measuring the acoustic signal travel time delay between the AUV and the seafloor transponder [16]. The positioning system generally arranges more than three datum arrays on the seabed, observes the signal delay between the detected AUV and each array element and solves the position of the target. This requires the precise calibration of the seafloor transponder by a survey vessel, a process that often takes a long time [29,30].

As shown in Figure 1, the mathematical equations of LBL are based on the spherical case. According to this, we have the following equation [30,31]:

$$L_i = ct_i + \varepsilon_L \quad (i = 1, \ldots, N) \tag{1}$$

where $L$ is the distance observation from the transducer to the transponder; $c$ is the sound velocity; $t$ is the travel time of the acoustic signal; $\varepsilon_L$ is the equivalent ranging error; $N$ is the number of datum points. When there are three or more observations, the position of the AUV can be calculated.

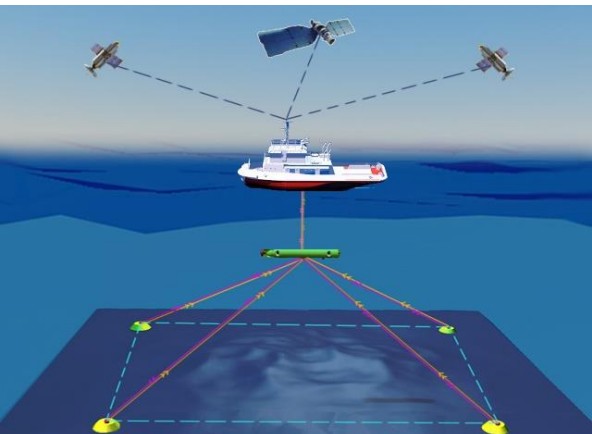

**Figure 1.** The principle of navigation and positioning of the AUV based on two models. (The pink and orange arrows indicate the signal transmission and return.)

Then, the coordinates of the transducer carried on the AUV are set to $X$ ($X$, $Y$, $Z$), and the coordinates of the fixed known datum points on the seabed are $(x_i, y_i, z_i)$. The following relations can be formulated:

$$(X - x_i)^2 + (Y - y_i)^2 + (Z - z_i)^2 = L_i^2 \quad i = 1, \ldots, N \tag{2}$$

This equation clearly indicates that the AUV is on the surface of a sphere centered around the corresponding transponder position and with a radius equal to $L_i$.

It is generally assumed that the $Z$-coordinate of the AUV is known (as provided by a depth gauge or altimeter carried on the AUV). Similarly, it is usually assumed that the datum points deployed on the seabed are at the same depth, i.e., $z_1 = z_2 = \ldots = z_N$. The problem of finding the position of the AUV is then transformed into a 2D problem. We have the equations

$$(X - x_i)^2 + (Y - y_i)^2 = r_i^2 \quad i = 1, \ldots, N \tag{3}$$

where the term

$$r_i^2 = L_i^2 - (Z - z_i)^2 \quad i = 1, \ldots, N \tag{4}$$

is obtained by the measured travel time and the known vertical distance between the AUV and the transponders. When there are four transponders, the following linear relationship can be obtained:

$$AX = R - l \tag{5}$$

where

$$A = \begin{bmatrix} x_1 - x_2 & y_1 - y_2 \\ x_2 - x_3 & y_2 - y_3 \\ x_3 - x_4 & y_3 - y_4 \\ x_4 - x_1 & y_4 - y_1 \end{bmatrix}; \quad R = \frac{1}{2}\begin{bmatrix} r_2^2 - r_1^2 \\ r_3^2 - r_2^2 \\ r_4^2 - r_3^2 \\ r_1^2 - r_4^2 \end{bmatrix}; \quad l = \frac{1}{2}\begin{bmatrix} (x_2^2 + y_2^2) - (x_1^2 + y_1^2) \\ (x_3^2 + y_3^2) - (x_2^2 + y_2^2) \\ (x_4^2 + y_4^2) - (x_3^2 + y_3^2) \\ (x_1^2 + y_1^2) - (x_4^2 + y_4^2) \end{bmatrix} \tag{6}$$

The position $X(X, Y)$ of the AUV can be estimated with the least squares method:

$$\hat{X} = (A^T A^{-1}) A^T (R - l) \tag{7}$$

The full differentiation of the range equation is obtained [32]:

$$(X - x_i)(dX - dx_i) + (Y - y_i)(dY - dy_i) = ct_i(cdt_i + t_i dc) \tag{8}$$

where $dc$ is the sound velocity error; $dt_i$ is the time error, which is caused by the clock error and pulse front measurement error (related to the signal/noise ratio); and $dx_i$, $dy_i$ and $dz_i$ are the base datum position errors. The sound velocity error is a very important error that affects the positioning accuracy, and it exists in both time and space characteristics, so the sound velocity environment must be accurately measured before each experiment [21,33].

The advantage of the LBL system is that it has a large tracking and positioning range and can effectively position the target within a range of tens of kilometers with high accuracy. Under the strict calibration of the underwater base array, centimeter–decimeter-level positioning can be realized after error correction. However, the disadvantages are also obvious, i.e., the underwater acoustic base array needs to be strictly calibrated and it is difficult to deploy and recover, not flexible enough and costly.

Similarly to Figure 1, the USBL positioning principle is based on measuring the travel time between the transducer and the transponder based on a known sound velocity profile. The difference is that USBL requires knowledge of the incident angle, which is obtained by measuring the phase difference. The following relation exists:

$$X = G(c, t, \varphi, f, d) + \varepsilon \tag{9}$$

where **G** is a realization function; $\varphi$ is the phase difference; $\phi$ is the incident angle; $\varepsilon$ is the observation error. A more detailed system of equations for the derivation of USBL can be found in [14].

The USBL system is used more frequently than the LBL system in actual marine surveys and exploration engineering. We believe that there are two main reasons for this: one is that the USBL is more flexible and less costly; and the other is that, for most exploration projects, the navigation and positioning does not require the level of accuracy provided by the LBL, e.g., navigation of ROVs (Remotely Operated Vehicles) and HOVs (Human-Occupied Vehicles). Combining the two factors, it can be concluded that the USBL is more cost-effective for deep and distant sea operations. However, in some application scenarios, such as fine topographic measurements of the seafloor using AUVs, which require high-precision navigation and positioning information, the integrated navigation system assisted by the LBL system can be applied.

### 2.2. LBL/SINS/DVL Integrated Navigation System Mode

Each sensor has its uniqueness and limitations, and in order to improve the accuracy of AUV navigation and positioning, multiple sensors' information needs to be fused. Currently, the shallow coupling mode is the most commonly used for underwater navigation systems. This model refers to integrated navigation systems in which the subsystems do not affect each other but only use the observed information from each navigation system to estimate the state and obtain optimal estimates of the navigation parameters. The schematic block diagram of the loosely shallow coupling mode based on LBL/SINS/DVL is shown in Figure 2.

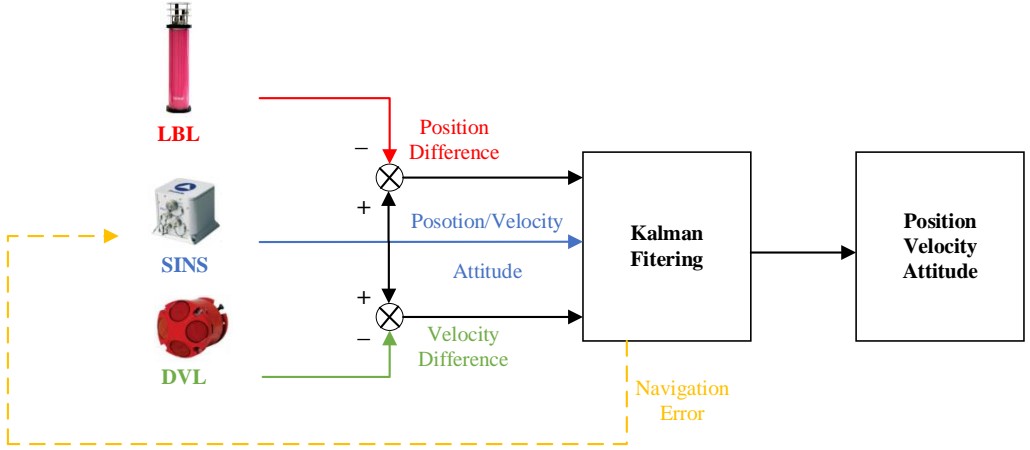

**Figure 2.** Block diagram of shallow coupling mode.

The state equation of the system is [26]

$$\dot{X} = FX + GW \tag{10}$$

where $X$ is the state variable; $F$ is the state patent matrix; $G$ is the process noise transfer matrix; $W$ is the system noise. $X$ is defined as

$$X = [\delta V_E \; \delta V_N \; \delta V_U \; \phi_E \; \phi_N \; \phi_U \; \delta L \; \delta \lambda \; \delta h \\ \nabla_{bx} \; \nabla_{by} \; \nabla_{bz} \; \varepsilon_{bx} \; \varepsilon_{by} \; \varepsilon_{bz}]^T \tag{11}$$

where $\delta V_E$, $\delta V_N$, $\delta V_U$ are the velocity errors; $\phi_E$, $\phi_N$, $\phi_U$ are the angles of misalignment; $\delta L$, $\delta \lambda$, $\delta h$ are the positional errors; $\nabla_{bx}$, $\nabla_{by}$, $\nabla_{bz}$ are the accelerometer zero bias; $\varepsilon_{bx}$, $\varepsilon_{by}$, $\varepsilon_{bz}$ are the gyro drift error; $F$ can be determined from the SINS error equation.

The measurement equation of the system is

$$Z = \begin{bmatrix} P_{SINS} - P_{LBL} \\ V_{SINS} - V_{DVL} \end{bmatrix} = HX + V \tag{12}$$

where $Z$ is the observation measurement; $P_{SINS}$ is the position information output by the SINS; $P_{LBL}$ is the position information output by the LBL; $V_{SINS}$ is the velocity information output by the SINS; $V_{DVL}$ is the velocity information output by the DVL; $V$ is the observation noise vector; and $H$ is the observation matrix.

$$H = \begin{bmatrix} 1 & 0 & 0 & 0 & 0 & 0 & 0 & 1 & 0 & 0 & 0 & 0 & 0 & 0 & 0 \\ 0 & 1 & 0 & 0 & 0 & 0 & 0 & 0 & 1 & 0 & 0 & 0 & 0 & 0 & 0 \\ 0 & 0 & 1 & 0 & 0 & 0 & 0 & 0 & 0 & 1 & 0 & 0 & 0 & 0 & 0 \end{bmatrix} \tag{13}$$

### 2.3. Three Navigation and Positioning Modes for AUV

The acoustic equipment carried on the AUV is available in both transducer and transponder modes. This means that the AUV operates in both USBL and LBL modes. If multiple seafloor beacons are not deployed, LBL positioning is not possible, in which case the system will adopt "Mode I". At the same time, the onboard SINS, OCTANS and DGPS motion sensors work together to correct the attitude and position of the USBL sensor, making its positioning more accurate. In addition, when the position of the AUV provided by the USBL is obtained, the position information is fused with the information from the SINS, DVL and DG (Depth Gauge) to form the navigation parameters of the AUV, which is "Mode III".

The LBL works properly and provides more accurate position information when multiple datums are placed on the seabed in the early stages. The AUV is in the "Mode II" navigation mode at this time. Likewise, after obtaining the position of the beacon on the AUV, the results are fed to the integrated navigation system and fused with the results of the SINS, DVL and DG to obtain the parameters required for AUV navigation. The conversion of the three navigation and positioning modes of the AUV is shown in Figure 3.

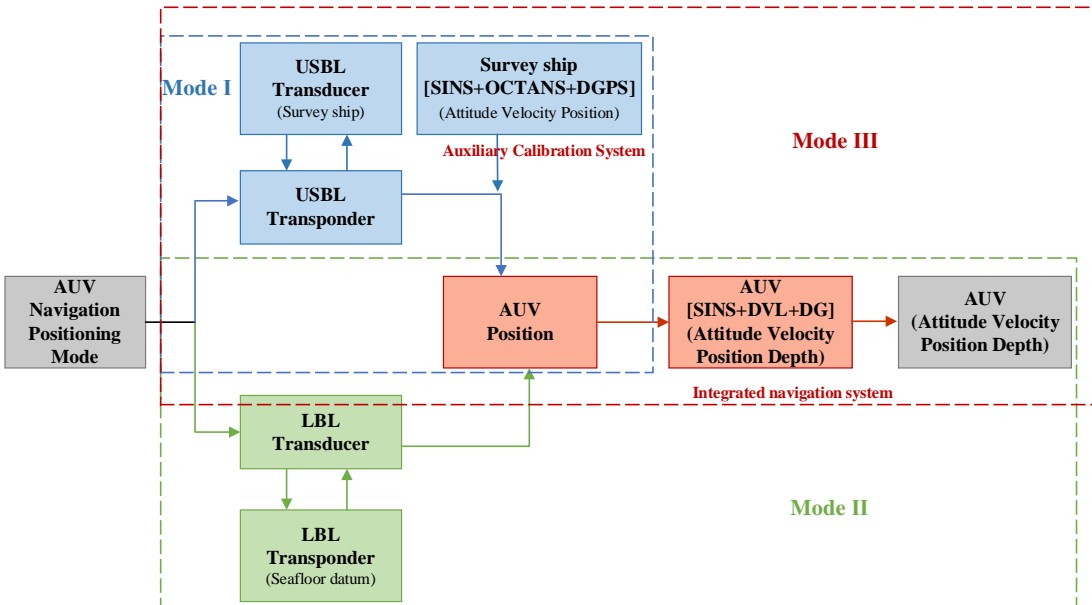

**Figure 3.** The navigation and positioning mode conversion of AUV.

It should be noted that when using both modes simultaneously, it is necessary to set the trigger periods of different frequencies to ensure that navigation and positioning data can be obtained in both modes. For example, the trigger period of the USBL signal is set to

36 s, while the LBL signal is set to 15 s. This setting should also take into account the depth of the water and the time taken for the acoustic signal to travel.

## 3. Experiment and Analysis

### 3.1. Description of the Background to the Experiment

The area where this experiment was conducted was a specific sea area in the South China Sea, with a maximum depth of approximately 4500 and a relatively flat seabed terrain. The weather conditions on the day of the AUV dive experiment were favorable and the sea state was Level 2.

The survey ship carried various sensors for the experiments, such as sound velocity profilers and the "Posidonia II" ultra-short baseline positioning system from IXBIUE, etc. It should be noted that the "Posidonia II" has nominal positioning accuracy of $0.2\% \times D$, where $D$ is the slope distance. The "Explorer 6000" AUV from Canada's ISE is a modular product that is tailored to the user's operational requirements. The product is highly scalable and can be easily configured to meet new needs. This configuration of the ISE Explorer AUV is capable of 2.5 m/s max velocity, with a depth rating of 6000 m. It is dynamically controlled in depth, roll, pitch, yaw and velocity. For survey missions, the vehicle has a nominal cruise velocity of 1.5 m/s. The sensors installed on the Explorer include the following: Obstacle Avoidance Sonar, "Ramses-6000" Beacon Transducer, Acoustic Modem Transducer, Depth Sensor (0.01%), Sound Velocity Sensor ($\pm 0.05$ m/s), DVL ($\pm 0.5\%$ $V$ (Velocity) $\pm 0.5$ cm/s), GPS, PHINS, etc. It is also important to note that the "Ramses 6000" has range accuracy of less than 0.05 m and positioning accuracy of less than 0.1 m. The PHINS system is an INS. PHINS uses 0.01 deg/h fiber optic gyroscopes and includes a Kalman filter. The Kalman filter is designed to optimally merge information from the GPS, LBL (or USBL), DVL and depth sensor with inertial data from the IMU.

The entire mission was set up before the experiment began, including the deployment of personnel and the design of the AUV navigation path. Before the AUV enters the water, a safety experiment was carried out, mainly to check the integrity of the hull and that the various sensors were working properly to ensure its safety.

The robotic arm and crew worked together to lower the AUV into the sea and move it freely under telemetry control. The operations at the site are shown in Figure 4. When the AUV moved away from the mothership and reached a safe position, a deep dive command was given via the surface control computer on the mothership. The AUV spiraled down to a specified depth at a set velocity and dive angle. Acoustic communication sensors called the AUV at set intervals, so the status of the AUV could be viewed in "real time" by the surface control computer. Note that the USBL on the mothership was already operational when the AUV started to dive, allowing us to see the 3D coordinates of the AUV position. The PHINS and LBL also became operational while the AUV was diving, but the DVL only became operational when the AUV was less than 80 m above the seafloor.

### 3.2. Deep-Sea Navigation and Positioning Experiments and Analyses of AUV

Acoustic signal propagation in water is affected by the seawater CTD (Conductivity, Temperature, Depth), so, when measuring the position parameters, the positioning accuracy is sensitive to the SVP (Sound Velocity Profile). Therefore, the SVP and CTD were measured in the experimental area prior to the AUV dive, as shown in Figure 5. The SVP was used for acoustic positioning and the CTD was used for the solution of the AUV navigation parameters.

As can be seen in Figure 5, the maximum observed water depth was about 4200 m, the maximum sound velocity reached about 1541 m/s, and the minimum sound velocity decayed to about 1483 m/s. It is worth noting that although the observations were made at the same time and at the same position, there was a small difference between the sound velocity calculated from the CTD observations and the results obtained from direct measurements of the sound velocity. There was still a sound velocity error of about

2 m. Based on the law of sound refraction, this error could result in centimeter-level positioning errors.

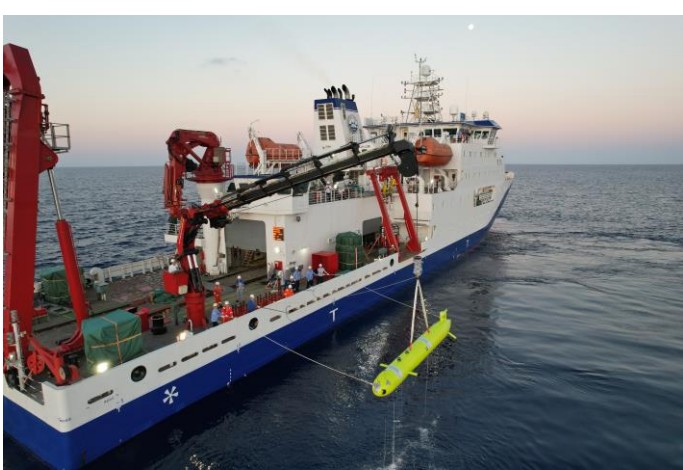

**Figure 4.** Situation at the site of the AUV deployment operation.

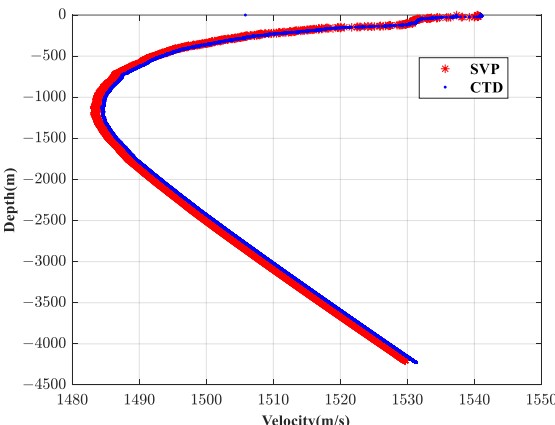

**Figure 5.** SVP and CTD.

The LBL system can be used for position calibration, primarily by setting datum points on the seabed, and the typical number is four. Figure 6 shows the layout of the datum points and accurately locates each one through circular calibration. The radius of the circle is equal to the water depth. The symbol "+" indicates the coordinates of the solved datum points. The calculation results show that the positioning accuracy of the four points reaches $0.5‰ \times D$. The integer parts of the latitude and longitude have been removed for ease of reading, and the same will be done below.

The PHINS sensor carried on the AUV records information about its attitude, and Figure 7 shows the variation in the AUV attitude in three degrees of freedom throughout the dive.

The first stage of the pitch is almost always negative and ranges from $0°$ to $-45°$, indicating that the AUV is diving; in the second stage, the pitch fluctuates around $0°$, indicating that the AUV is maintaining horizontal navigation near the bottom. From the viewpoint of the roll, the floating range is $(-10°, 10°)$, and it changes only when the rudder turns during the AUV's steering in the dive phase and near-bottom phase. The heading changes very frequently and is only maintained during directional navigation.

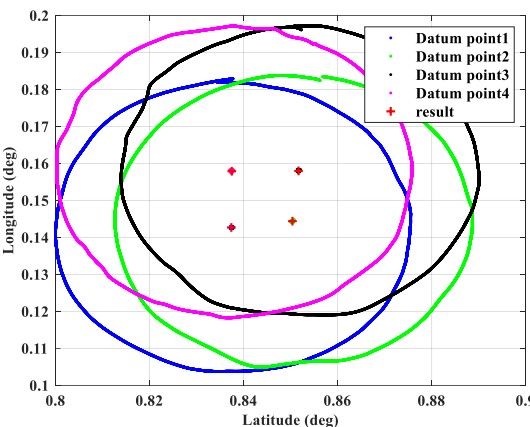

**Figure 6.** Array of four datum points.

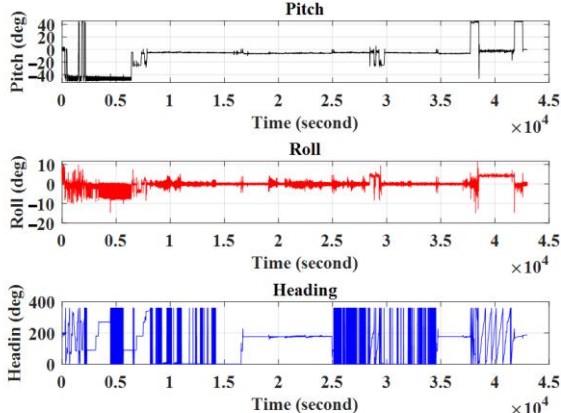

**Figure 7.** Attitude changes during AUV diving.

The forward velocity information for the AUV combines velocity data from PHINS and DVL. As shown in Figure 8, in the first stage, PHINS achieved good results in data output after GPS correction over the water surface, and the velocity information comes solely from PHINS; in the second stage, the velocity data are smoother, and, at this time, the velocity information of PHINS is calibrated by DVL during the diving process near the seabed. In the third stage, when the AUV starts to float upwards, the DVL is no longer effective and the velocity of PHINS is not corrected, so the results no longer have reference value.

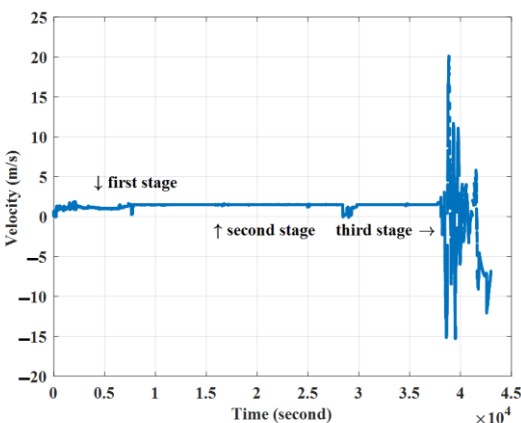

**Figure 8.** Forward velocity during AUV diving.

The depth and altitude information shown in Figure 9 was obtained from the depth gauge and altimeter sensors carried on the AUV. The AUV was tested in a short hover at a depth of 800 m. At a time of approximately $2.8 \times 10^4$ s, the AUV underwent a brief ascent to perform a low-speed cruise test. The altimeter starts working when the AUV is less than 300 m from the seabed, which can more accurately display the AUV and allows the pilot to adjust the hull for safety at any time.

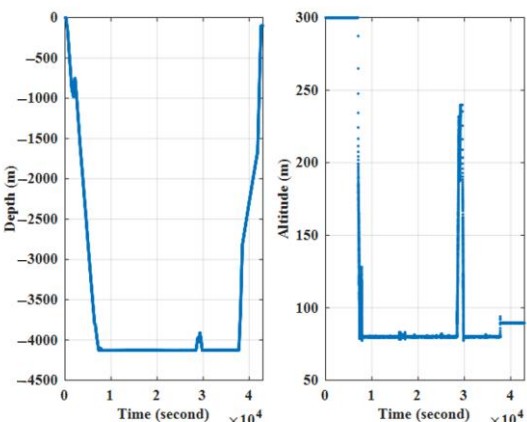

**Figure 9.** Depth and altitude information during AUV diving.

To evaluate the accuracy of the navigation and positioning of the AUV, the trajectory of the AUV after the dive is given in Figure 10 and the position is obtained from the "Posidonia II" USBL on the mothership. It is important to note that we have simplified "Mode I" because the AUV navigation cannot be performed in both "Mode I" and "Mode II" at the same time. Here, the onboard USBL only locates the AUV and does not perform combined multi-sensor navigation, which is the first half of the process in "Mode I".

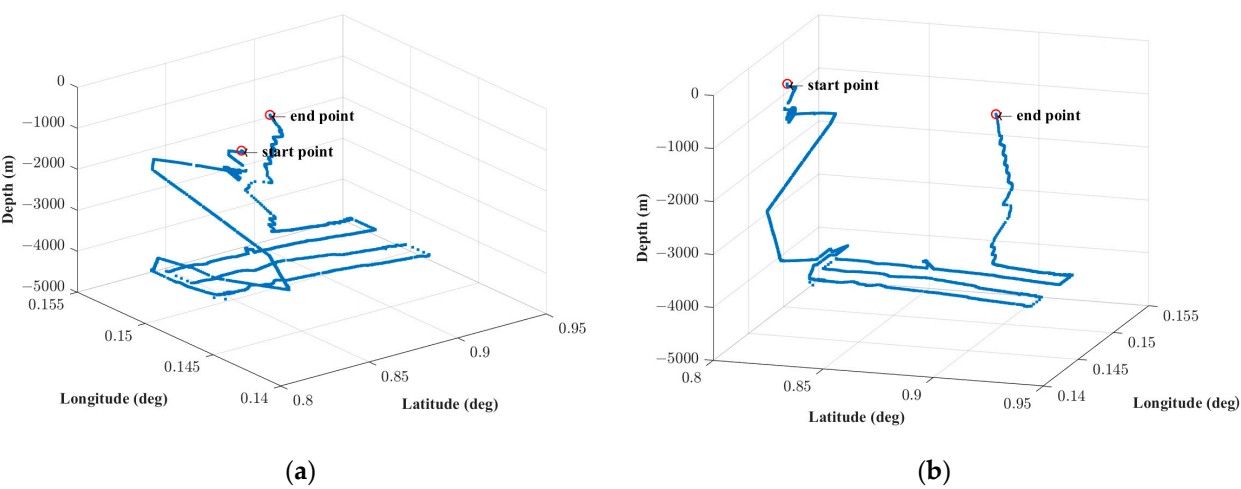

(**a**)  (**b**)

**Figure 10.** AUV trajectory from "Mode I". (**a**) Viewpoint 1; (**b**) Viewpoint 2.

It can be seen from Figure 10 that the AUV has an irregular trajectory at the start of the dive. The low positioning accuracy of the USBL for shallow-water targets is due to the fact that the hydrological environment in shallow water can adversely affect the signal propagation of the USBL system; in addition, the algorithms are not able to deal with acoustic signals when they propagate horizontally. Similarly, the accuracy of USBL for deep-water targets is not satisfactory due to the influence of factors such as sound velocity variations and travel time delays. The results show that the trajectory points of the AUV are not smooth and there are obvious jump points. The "Posidonia II" claims that it can

achieve positioning accuracy of 0.2% × *D*, but it is difficult to assess the actual navigation accuracy of the AUV from this experiment alone.

To compare the navigation accuracy of the AUV in the two modes, the navigation trajectories of the AUV in the two modes are shown in Figure 11, where the coordinates of the four datums are calibrated as in Figure 6. The AUV dive workflow is briefly described below: the AUV is submerged within the datum array; it starts to navigate according to the pre-set survey line after approaching the seabed; and it climbs to the surface to retrieve itself after completing its work at a certain point outside the datum array.

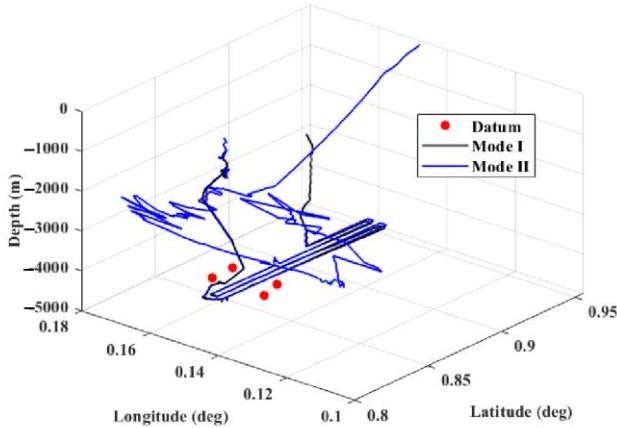

**Figure 11.** AUV trajectory in both modes.

The results show that when the AUV is in the third stage of the climb, the results of "Mode II" differ significantly from the results of "Mode I". Combined with the field work, there are two main reasons for this. The first reason is that the position is far from the datum array when the AUV is in the climbing stage, and the position information provided by LBL is inaccurate at this time. Moreover, the SINS is not forced to trust the LBL, i.e., if the LBL is inaccurate, SINS will reject the position correction from LBL, and the DVL will stop working at this time, resulting in a large deviation in the SINS navigation results. The second reason is that the AUV climbs with its belly facing the LBL datum array, which also causes the LBL positioning to fail. The combination of these two factors leads to the failure of "Mode II" navigation during the climb phase of the AUV.

After deleting the data from the AUV climb phase, the results are obtained as shown in Figure 12. In general, "Mode II" is significantly better than "Mode I" both inside and outside the datum array, mainly due to the assistance provided by the DVL in navigating near the bottom. In addition, the accuracy of "Mode I" is significantly reduced when the AUV is sailed outside the datum array due to the position of the ship on the water surface, which is directly above the datum array. The reason for this is related to the USBL positioning error analysis, i.e., changes in range and observation angle lead to increased positioning errors. A magnified view of the trajectory of a survey line is shown in Figure 13.

From a portion of the AUV near-bottom trajectory, the maximum position error in both modes is approximately 80 m. In addition, the other two locations have position errors between 30 m and 50 m. Therefore, single positioning using USBL alone is not reliable in long-range and deep-sea situations.

The purpose of this experiment was to compare the accuracy of AUV navigation in "Mode I" and "Mode II". The reason that we do not compare the accuracy of "Mode II" and "Mode III" in this paper is that these two modes cannot be considered in navigation experiments at the same time. In general, considering the cost efficiency and other issues, when "Mode II" cannot be implemented, the accuracy of "Mode III" can also enable the AUV to carry out relevant work.

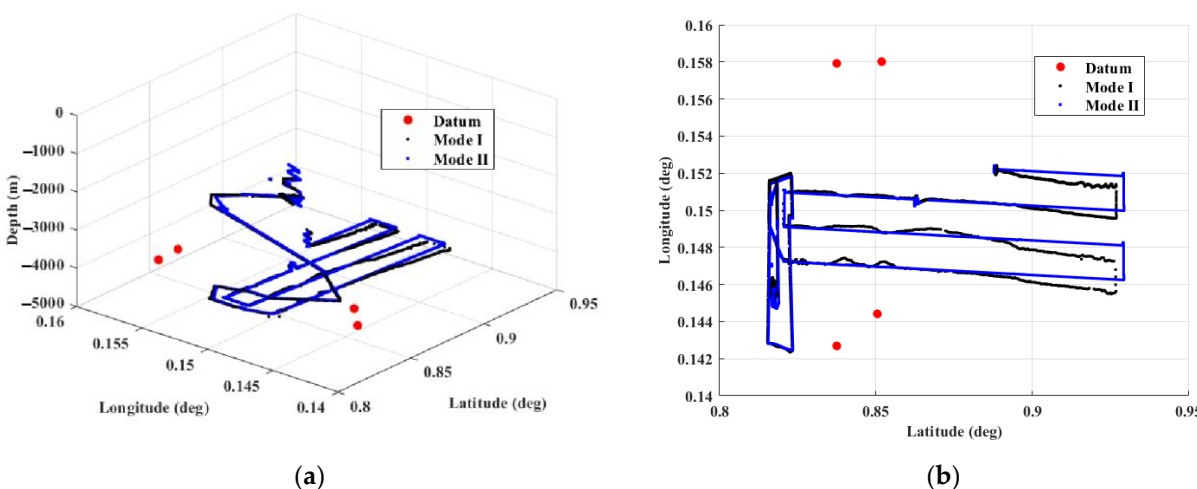

(**a**)  (**b**)

**Figure 12.** AUV trajectory in both modes (data do not include AUV climb phase). (**a**) Front view; (**b**) top view.

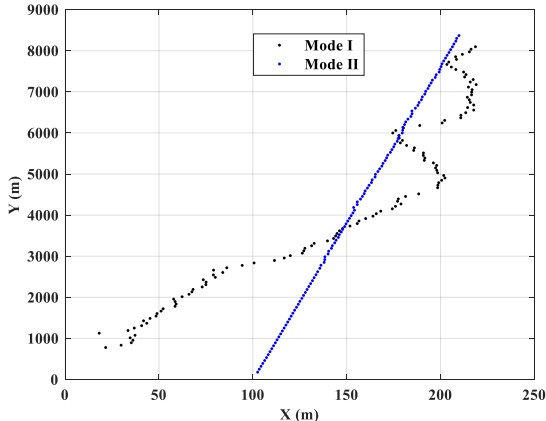

**Figure 13.** Part of the near-bottom trajectory of an AUV.

## 4. Discussion

In line with the above discussion, AUVs operating in deep-sea scenarios face significant challenges, particularly in terms of navigation and positioning accuracy. The problem of the SVP is the primary problem facing high-precision positioning. It has significant spatial and temporal characteristics, and this variation is of a very considerable order of magnitude relative to the sound velocity itself. This results in oceanic acoustic refraction effects being more severe than those of electromagnetic waves. As shown in Figure 5, even when the SVP and CTD are used for simultaneous observations, there is still a sound velocity error of about 2 m, which has a non-negligible impact on high-precision positioning. Some experts are starting to study the effect of the SVP error on seafloor precision positioning and its parametric estimation, which is used to solve the problem of the temporal and spatial representative error of velocity, and we hope that the results of this paper will be of reference value for this type of study.

Further, the prerequisite for high accuracy is a highly reliable sensor, followed by a good navigation and positioning algorithm. This is difficult to achieve with a single sensor, as can be seen from the USBL accuracy in Figure 13. In the case of this AUV, the accuracy of the USBL positioning mode is higher when the AUV is within a 60° observation range below the ship; meanwhile, the positioning accuracy decreases with increasing range and observation angle, i.e., the positioning error reaches 80 m at a 4000 m depth. It should be added that, according to the experimental results of other works, the accuracy of the USBL system is usually in the range of 2‰~5‰ $\times D$. However, the USBL system also needs other

sensors to help to improve the localization accuracy, as with Mode I. Based on this, the USBL positioning accuracy can reach $1‰ \times D$.

In addition, the reliability of the LBL system's accuracy is unquestionable, provided that the reliable SVP mentioned above is still available. However, the use of this system is complex, e.g., as seen in Figure 6, datum arrays are costly to deploy and recover, they are precision repeaters, and they can significantly affect the positional accuracy of the AUV. The navigation accuracy inside and outside the array is high when the LBL/SINS/DVL mode is used; when the AUV is far away from the datum array in the process of climbing to the surface, the LBL cannot provide accurate position calibration, resulting in large deviations in the SINS results. Of course, the prerequisite for LBL to achieve high-precision positioning is a high-precision seafloor datum array and error correction theory calibration, as shown in Figure 6.

Of note, DG data are used in integrated navigation systems because acoustic positioning systems are less accurate in the elevation direction than in the horizontal direction. This is similar to GPS and is associated with the asymmetry of the data observations. DG data can provide strong constraints in the elevation direction that can be used to improve the positioning accuracy.

In summary, the motivation behind this work was to gain a practical understanding of the needs and identify the problems in deep-sea navigation and positioning research. All research is based on practical applications, and the aim of this deep-sea AUV mission is to detect the refined topography of the deep-sea to provide maps for manned submersibles to ensure safety. However, the navigation and positioning mode to be used in practical engineering is under discussion, and an increase in accuracy will inevitably lead to an increase in cost. A feasible way to improve the accuracy of the algorithm is with existing equipment. In addition, most papers in the field focus on simulation (and provide simulation results), but there are some problems that cannot be estimated in advance and therefore simulation experiments are inherently flawed.

## 5. Conclusions

The harsh environment of the deep sea makes the navigation and positioning of submarines critical to their safety and data reliability. An acoustic positioning system is currently the best method for underwater position calibration and integrated navigation. However, there has been little discussion of the simultaneous use of both USBL and LBL modes on an AUV, especially in deep-sea scenarios. The process of comparing the two modes is also a process of the mutual verification of their accuracy.

This paper analyzes and discusses the navigation and positioning of AUVs in deep-sea scenarios and compares the trajectories of an AUV in the two modes of USBL and LBL. First, the SVP and CTD data were observed simultaneously; second, accurate datum array coordinates were obtained using circular calibration by USBL. These two preliminary steps are critical to positioning. In the experiments, we found that the positioning accuracy of USBL for long-range and large-angle targets decreased, and this was more obvious on the sea surface. However, based on the cost and accuracy analysis, the navigation mode based on USBL position correction is still the most commonly used at present. In the case of the studied AUV, LBL can provide decimeter-level positional accuracy within the datum array, and, in conjunction with sensors such as SINS and DVL, it can provide meter-level navigation services; however, outside the datum array and when the DVL loses its function, it can cause the results of SINS to drift.

Based on the above discussion, we believe that if the LBL position calibration mode is adopted, the datum array should be reasonably arranged to allow the AUV to work within the array. If the USBL position calibration mode is used, coordination between the ship and AUV should be carried out to reduce the impact of the associated errors. Although, in principle, the accuracy of the LBL system is obviously higher, the actual accuracy of the two systems will also vary according to different conditions, such as the manufacturer and price.

The research of this paper had certain limitations that should be addressed. For example, the experiment did not directly compare the accuracy of "Mode II" and "Mode III", which represents a design flaw. We hope to supplement the ideas that have not been realized in this paper and achieve good results in future research. The low-cost deep-sea rapid positioning algorithm (without SVP observation) is also under research. We will propose a new algorithm and verify it with experiments.

As a supplementary discussion, the following problems identified through this experiment and data processing are worthy of study: (i) the time and economic cost of repeated observations of the SVP is very high; (ii) the navigation and positioning accuracy is very poor during the descent of the AUV, especially in the elevation direction; (iii) the positioning accuracy of the LBL system decreases significantly outside the datum array; (iv) when the AUV is steering, the communication and positioning effect is so poor that the ship cannot receive the data feedback at that moment; (v) when the AUV operates close to the seafloor, the ship must follow the AUV on the surface due to the long observation distance; otherwise, the communication and positioning are poor. We hope that, through this research, we can identify the problems, discuss them and obtain new research results.

**Author Contributions:** Conceptualization, Y.L. and Y.S.; Data curation, X.W.; Formal analysis, B.L.; Investigation, B.L.; Methodology, Y.L.; Project administration, L.Y.; Validation, Y.L.; Writing—original draft, Y.L.; Writing—review and editing, Y.S. All authors have read and agreed to the published version of the manuscript.

**Funding:** This research was funded by the National Key Research and Development Program of China (No. 2021YFC2803005; 2021YFC2801605) and Shandong Provincial Natural Science Foundation, China (No. ZR2022MD03).

**Data Availability Statement:** The datasets analyzed in this study are managed by the National Deep-Sea Base Management Center.

**Conflicts of Interest:** The authors declare no conflicts of interest.

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
