# Peer review of "Experimental Analysis of Deep-Sea AUV Based on Multi-Sensor Integrated Navigation and Positioning"

_remotesensing, doi:10.3390/rs16010199_

Round 1
Reviewer 1 Report
Comments and Suggestions for Authors
Manuscript, entitled „Experimental Analysis of Deep-sea AUV Based on Multi-sensor Integrated Navigation and Positioning„. The article reports that the navigational accuracy inside and outside the datum array is high when using the LBL/SINS/DVL mode; if the AUV is far from the datum array when climbing to the surface, the LBL cannot provide accurate position calibration while the DVL fails, resulting in large deviations in the SINS results. There are shortcomings and modifications that should be included in order to enhance the final manuscript for the readers.
Abstract:
1- Line 13. Please add the full name of the AUV as well as check the all abbreviations which were mentioned for the first time. 2- Line 17. Please replace the paper by the study. 3- Lines 19. The accuracy of USBL and LBL/SINS/DVL modes. For what?4- The conclusion of abstract should be added with the benefit of using Multi-sensor Integrated Navigation and Positioning.
Keywords:
5- Keywords should be arranged alphabetic?
Introduction:
6- Introduction was good written
7- From 2. Multi-sensor Integrated navigation system model to 4.2 Deep-sea navigation and positioning experiments and analyses of AUV. the authors should divide the manuscript to parts materials and methods as well as the results.
Discussion
8- Discussion is very weak. It is needed to write again and support by previous studies to compare this work. The discussion should cover the results of the manuscript. I suggest the authors integrate the results and discussion together9- Please, write the practical applications of your work in a separate section, before the conclusions and provide your good perspectives?
Conclusion
10- Please write about the limitations of this work in details? Comments on the Quality of English LanguageMinor editing of English language required
Author Response
Thank you very much for your encouragement and comments. We have revised the manuscript carefully and responded point by point to the comments as below. Our revisions are highlighted in our manuscript using the "Track Changes" function.

Reviewer 2 Report
Comments and Suggestions for Authors - Conclusions should be expanded by deductions in more details and suggestions for future works- There are some errors in printing and making
- References have not been updated- Need to compare with other methods Comments on the Quality of English Language
Author Response

(The authors gave the same response as above.)

Reviewer 3 Report
Comments and Suggestions for Authors
Pros: experimental data from in-water trials is presented. This alone makes this work worthy of being published, since most papers in this field focus on simulations (and provide simulated results), leaving no room for experimental in-water testing (and data from physical experiments).
Cons: there is nothing novel about this work. An evaluation of commercially available positioning methods for underwater navigation (LBL vs USBL) is presented.
The main areas that need some improvements are the following:
(1) only one specific AUV is used to collect data to compare USBL vs LBL. Is that enough to consider the conclusion applicable to all USBL and LBL systems? If not, the wording in the conclusion and the discussion should be revised.
(2) It is unclear why Mode III could not be compared to Mode I and II. Although, there is a paragraph where the author tries to explain why, the reasoning does not seem sufficient. Why couldn't the experiment be replicated to just compare Mode II and Mode III, if Mode I and Mode III cannot be run during the same experiment?
(3) The discussion and the conclusion contain an abundance of vague terms (reasonable observation range, far away, decreases significantly, accurate position information, high-precision navigation, etc.) that need to be quantified based on the data collected.
Author Response

(The authors gave the same response as above.)

Reviewer 4 Report
Comments and Suggestions for Authors
The topic of the manuscript is interesting and dedicated to the current problem of ensuring the accuracy of AUV positioning when operating at great depths. To ensure maximum positioning accuracy on board the AUV, it is proposed to use various combinations of available navigation equipment for various operating situations. Three combinations are being considered:
Mode I: USBL APS only
Mode II: LBL APS + on-board dead reckoning system sensors (SINS+DVL+DG)
Mode III: USBL APS + on-board dead reckoning system sensors (SINS+DVL+DG)
Depending on the depth of the AUV dive, its angular parameters, distance from the bottom and position relative to the base array of the bottom transponder beacons, it is proposed to choose one of the above-mentioned modes for operation.
The title of the manuscript and its abstract promise the reader some new information on the topic of deep-sea diving of AUVs. After reading the manuscript, you feel deceived in your expectations. The described navigation modes I, II and III are standard and their characteristics are well known. The recommendations given in the manuscript on the conditions for using these modes are also well known to specialists involved in the development and operation of AUVs. The experiment described in Section 4 is a standard deep-sea survey mission for AUVs, and the results presented are also typical and predictable. Thus, the manuscript does not provide any new information regarding the organization of work with AUVs.
From the reviewer’s point of view, a manuscript on a similar topic would make sense if it contained a description of the methodology and algorithms for automatically switching the AUV onboard navigation system between modes I, II, III in order to obtain maximum positioning accuracy depending on the developing navigation situation. Publication of the manuscript is inappropriate in its current form.
Spotted typos.
1) There is an error in the numbering of the manuscript sections. The “Discussion” and “Conclusion” sections should be numbered 5 and 6 respectively.
2) Line 366: instead of the phrase “clime phase” there should be “climb phase”.
Author Response

(The authors gave the same response as above.)

Round 2
Reviewer 1 Report
Comments and Suggestions for Authors
The manuscript was improved by the authors according to my comments. The manuscript can be accepted for publication.
Author Response
Thank you very much for your encouragement and recognition, so that I can be more confident in my research results. More importantly, my research results can be seen and discussed by others. Thank you again for the time and effort you put into my manuscript.

Reviewer 4 Report
Comments and Suggestions for Authors
The reviewer highly appreciates the amount of changes that the authors made to the original version of the manuscript. Particularly significant changes were made to the “Discussion” and “Conclusions” sections, where the authors tried to substantiate in more detail the need and significance of their experiments on the use of several options for AUV deep-sea positioning. The reviewer also highly appreciates the efforts spent by the authors on carrying out the deep-sea experiment, since from his own experience he knows the difficulty of execution deep-sea work using AUVs.
Nevertheless, the reviewer believes that the manuscript has not undergone fundamental changes. The experiment with the simultaneous use of USBL and LBL systems is certainly interesting. However, the results obtained from comparing the positioning accuracies of these systems in the same deep-sea conditions are predictable and well known to specialists. To date, quite a lot of experimental and practical work has been carried out using AUVs at great depths. For example, searching and photographing of sunken submarines or collecting data on aircraft wreckage scattered over a large area of the bottom surface after crashes. As a result, there is currently a complete understanding of when and in which modes it is more convenient to use one or another system, depending on the required positioning accuracy, the size and depth of the area being surveyed, mobility requirements, etc. In this context, the manuscript provides little new or interesting information.
The reviewer believes that a manuscript on a similar topic would make sense if it contained a description of the methodology and algorithms for automatically switching the AUV onboard navigation system between modes I, II, III in order to obtain maximum positioning accuracy depending on the developing navigation situation. Or the manuscript would contain a description of methods and algorithms for increasing the accuracy of USBL or LBL systems.
Author Response
First of all, we would like to thank you for your encouragement, which shows that the changes we have made are worthwhile, and our aim is also to make the reader's understanding easier through better discussion, and it would be nice to get some new thinking. Below we respond to the other suggestions you made, which we try to summarise in two questions. Our revisions are highlighted in our manuscript using the "Track Changes" function.

Round 3
Reviewer 4 Report
Comments and Suggestions for Authors
Dear Colleagues! Thank you for your thorough responses to the reviewer's comments. The reviewer fully agrees with you that the manuscript should contain new algorithmic solutions and/or new experimental data. But this manuscript has neither one nor the other! You yourself know about the absence of a description of new algorithmic solutions in the manuscript. As new experimental data, you propose to consider the following:
1) the time and economic cost of repeated observations of the SVP is very high;
2) the navigation and positioning accuracy is very poor during the descent of the AUV, especially in the elevation direction;
3) the positioning accuracy of the LBL system decreases significantly outside the datum array;
4) when the AUV is steering, the communication and positioning effect is so poor that the ship cannot receive the data feedback at that moment;
5) when the AUV operates close to the seafloor, the ship must follow the AUV on the surface due to the long observation distance, otherwise communication and positioning are poor.
But all the listed problems 1-5 are well known to specialists! The reviewer has already spoken several times about this. The experimental data described in the manuscript in one form or another have already been obtained previously during experimental missions and the practical use of numerous AUVs. Specialists around the world have been working on solving these problems for a long time.
For these reasons, the reviewer stands by his original opinion: the manuscript should be subject to radical revision. As you said yourself, the manuscript should contain new algorithmic solutions and/or new (really new!) experimental data.
Author Response
Thanks for your comments.We have responded to your questions in the following document.
